# Coping with the COVID-19 pandemic: A cross-sectional study to investigate how mental health, lifestyle, and socio-demographic factors shape students' quality of life

Imad Bou-Hamad[1]☯*, Reem Hoteit[2]☯*, Sahar Hijazi[3], Dinah Ayna[4], Maya Romani[5], Christo El Morr[6]

1 Department of Business Information and Decision Systems, Suliman S. Olayan School of Business, American University of Beirut, Beirut, Lebanon, 2 Department of Internal Medicine, Clinical Research Institute, American University of Beirut, Beirut, Lebanon, 3 Faculty of Social Sciences, Lebanese University, Saida, Lebanon, 4 Faculty of Medicine, Department of Psychiatry, American University of Beirut Medical Center, Beirut, Lebanon, 5 Faculty of Medicine, Department of Family Medicine, American University of Beirut Medical Center, Beirut, Lebanon, 6 School of Health Policy and Management, York University, Toronto, Canada

☯ These authors contributed equally to this work.
* ib12@aub.edu.lb (IB-H); rah84@aub.edu.lb (RH)

**Data Availability Statement:** We made the data publicly available by depositing it in a data bank maintained by The American University of Beirut's

## Abstract

### Purpose

The high prevalence of COVID-19 has had an impact on the Quality of Life (QOL) of people across the world, particularly students. The purpose of this study was to investigate the social, lifestyle, and mental health aspects that are associated with QOL among university students in Lebanon.

### Methods

A cross-sectional study design was implemented using a convenience sampling approach. Data collection took place between November 2021 and February 2022, involving 329 undergraduate and graduate students from private and public universities. Quality of life was assessed using the Quality-of-Life Scale (QOLS). Descriptive statistics, Cronbach's alpha, and linear regression-based methods were used to analyze the association between QOL and socio-demographic, health-related, lifestyle, and mental health factors. The significance level for statistical analysis was predetermined at α = 0.05.

### Results

The study participants' average (SD) QOL score was 76.03 (15.6) with a Cronbach alpha of 0.911. QOL was positively associated with importance of religion in daily decisions (β = 6.40, p = 0.006), household income (β = 5.25, p = 0.017), general health ratings (β Excellent/poor = 23.52, p <0.001), access to private counseling (β = 4.05, p = 0.020), physical exercise (β = 6.67, p <0.001), and a healthy diet (β = 4.62, p = 0.026); and negatively

libraries. To access the data, use the following: DOI: http://hdl.handle.net/10938/24094.

**Funding:** The author(s) received no specific funding for this work.

**Competing interests:** The authors have declared that no competing interests exist.

associated with cigarette smoking (β increased = -6.25, p = 0.030), internet use (β ≥4 hours = -7.01, p = 0.005), depression (β = -0.56, p = 0.002) and stress (β = -0.93, p <0.001).

## Conclusion

In conclusion, this study reveals the key factors that positively and negatively influence students' quality of life (QOL). Factors such as religion, higher income, and a healthy diet improve QOL, while depression, stress, excessive internet use, and cigarette smoking negatively impact it. Universities should prioritize initiatives like physical activity promotion, affordable nutritious options, destigmatizing mental health, counseling services, and self-help interventions to support student well-being and enhance their QOL.

## Introduction

SARS-CoV-2 is a coronavirus that is highly infectious and contagious [1]. The Coronavirus disease (COVID-19) has been a serious worldwide health hazard since the WHO declared it a global pandemic on March 11, 2020 [2]. By 15 May 2023, COVID-19 has infected over 765 million individuals and caused over 6.9 million deaths worldwide. To limit the virus's rapid spread, worldwide and government health organizations implemented quarantine and lock-down measures such as flights' suspension, social distancing, public gatherings, enforced wearing of face mask, teleworking, home-schooling [3–5]. With the expansion of the pandemic, many educational institutions incorporated online activities, quickly changing the teaching-learning process [6] and disrupting the academic lives of many students throughout the world. Such changes, which are linked to social isolation during the pandemic, may have an impact on students' quality of life and perhaps lead to the escalation of psychological problems [7,8].

Quality of life (QOL) is defined "an individual's perception of their position in life in the context of the culture and value systems in which they live and in relation to their goals, expectations, standards and concerns" [9]. There has been a lot of research on how these aspects of a person vary over time, and whether changes in one (such as personality) induce changes in the other (like QOL) [10]. Students may face social, emotional, physical, and family challenges, which can have a negative impact on their academic performance and study abilities [11–13]. All of these elements may have an impact on university students' quality of life. Additionally, several predictors of QOL among university students have been identified, including gender, educational environment, years of study, depression, and chronic illness [14,15].

The COVID-19 pandemic has significantly negatively affected the quality of life of university students. A study conducted by Dos Santos et al. (2022) found that scores for quality of life, stress, and depression all worsened during the pandemic compared to pre-pandemic levels [16]. Additionally, the same study revealed that the pandemic led to increased physical inactivity among young people, and that female health students who lived independently and had not tested positive for COVID-19 experienced higher levels of stress during the pandemic [16]. Recent research has found a decline in QoL in general and, more particularly, in the anxiety/depression domain [17,18].

Another study conducted by Carpi et al. (2022) revealed high rates of poor sleep quality and insomnia among university students [19], and even after controlling for the influence of health-related factors and perceived stress, this study found a substantial association between sleep quality and both physical and mental health-related quality of life [19]. A study conducted in Saudi Arabia also found that smoking had a detrimental impact on the quality of life

of vocational students [20]. Besides, research conducted in Brazilian universities revealed that increased alcohol consumption during the COVID-19 pandemic was associated with a decline in quality of life, health satisfaction, and the meaning of life and that this negative impact was greater among women than men [21].

In Lebanon, the first verified case of COVID-19 was announced on February 21, 2020. As of 15 May 2023, the Lebanese Ministry of Public Health (MOPH) had confirmed over 1.2 million cases and over 10 900 deaths since the beginning of the COVID-19 pandemic [22]. Between 2020 and 2021, the government implemented various lockdowns in an attempt to flatten the curve [3,23]. Besides, several modeling studies have been conducted to assess the transmission of SARS-CoV-2 and evaluate the impact of COVID-19 vaccination [24–26]. One study, conducted in April 2021, predicted a significant increase in cases and deaths if schools and universities reopened, particularly considering the low vaccination rates of below 4% [26].

Despite various efforts to encourage vaccination, such as the national COVID-19 vaccination plan aiming to vaccinate 70% of the population by the end of 2022, Lebanon has fallen far short of achieving this goal by 2023 [27,28]. As of April 2023, the first-dose vaccine coverage stood at 50.4%, with second-dose coverage at 44.4%. Furthermore, only 27.6% of individuals who received the second dose went on to receive the third dose [29]. As of April 2023, new COVID-19 infections continue to occur in Lebanon, with an average of 100 cases reported daily.

Despite the global attention given to the impact of the COVID-19 pandemic on university students' quality of life, there is a significant gap in the literature regarding the situation in Lebanon. The current situation is unparalleled, as the country is grappling with an overwhelming crisis characterized by deteriorating financial and political stability that has plagued Lebanon for the past years, leading to a mass exodus of physicians and nurses [30,31]. It is noteworthy that the COVID-19 pandemic has exacerbated Lebanon's existing severe economic and financial crisis, which began in October 2019 [28]. In March 2023, the Lebanese lira experienced a devaluation of over 95%, making it one of the most severe economic and financial crises witnessed globally since the mid-19th century, according to the World Bank [32,33].

These factors have significantly impacted the daily lives of its citizens, including university students. While previous research has identified various factors affecting the QOL of university students, there is still a need to examine the specific impact of the pandemic and related lockdown measures on students' QOL in Lebanon. Therefore, this study aims to fill this research gap by investigating the various factors, including sociodemographic, health-related, lifestyle, and mental health factors, that affect university students' QOL in Lebanon during the pandemic. By providing insights into these factors, we hope to inform interventions and policies aimed at improving the well-being of university students in Lebanon.

## Materials and methods

### Study design and participants

This study employs a cross-sectional design to investigate the factors influencing the quality of life (QOL) among university students in Lebanon during the COVID-19 pandemic. An online survey was disseminated to undergraduate and graduate university students in Lebanon. The students were provided with a link to the survey along with a detailed study description via electronic platforms such as WhatsApp and email. To maximize participation and response rates, two reminder messages were sent to the participants within a two-week interval. The survey began with participants receiving a consent form that safeguarded their interests, provided them with relevant information regarding their rights and responsibilities, and assured the confidentiality of their information. On average, students took approximately 15–20 minutes to complete the survey.

Data was collected between November 2021 and February 2022, coinciding with the emergence and spread of the Omicron variant. The study sample included 329 undergraduate and graduate students who were 18 years of age or older and enrolled between Spring 2020–2021 and Fall 2021–2022 at the American University of Beirut (a private university) and the Lebanese University (a public university).

The questionnaire was administered in Arabic and English. All survey participants provided their written informed consent online before completing the survey. Considering the evolving nature of the pandemic and to minimize the spread of the virus, an online convenience sampling strategy was adopted. This sampling approach has been commonly employed in numerous COVID-19-related studies [3,34,35]. The participants received no monetary compensation, and the data was kept anonymous, and anonymity was maintained to ensure the confidentiality and reliability of data. This study was carried out in full accordance with the Declaration of Helsinki's guidelines for human subjects' research. The study received ethical approval from the American University of Beirut's Institutional Review Board (SBS-2021-0256) and the Research Ethics Board at York University in Canada (Certificate # e2021-327).

## Survey instrument

The survey instrument consisted of questions assessing students' socio-demographic characteristics, lifestyle practices, mental health factors and quality of life.

## Socio-demographic characteristics

Socio-demographic factors included age (continuous), gender (binary: male; female), household income (binary: ≤ USD 450; >USD 450), current program (categorical: undergraduate degree, certificate program, graduate program (Master of Arts (MA) or Master of Science (MSc)), PhD Program and Doctor of Medicine program (MD), relationship status (binary: not in a relationship, in a relationship), number of people living in the household (continuous), GPA status (categorical: no change, decreased, increased), importance of religion in daily decisions (binary: not important; important), conspiracy behind COVID-virus/vaccine (categorical: disapprove, neither approve nor disapprove and approve), adherence to COVID-19 preventive measures (binary: no; yes), infected with COVID-19 (binary: no; yes), access to private counseling (binary: no; yes).

## Lifestyle practices

Lifestyle practices during the pandemic included cigarette and shisha smoking (categorical: no practice, reduced and increased), alcohol intake (categorical: no practice, reduced and increased), physical activity (categorical: no practice, reduced and increased), sleeping hours (categorical: <7, 7–9, >9), internet usage (categorical: <1, [1–2], [2–3], [3–4] and ≥4), follow a healthy diet (binary: no; yes) and overall health (categorical: poor, fair, good, very good and excellent).

## Mental health

*Depression (PHQ-9; Kroenke, 2001).* The Patient Health Questionnaire (PHQ-9) [36] is a widely used and brief 9-item screening tool to detect symptoms of depression in community settings. The development of the PHQ-9 was based on the Diagnostic and Statistical Manual of Mental Disorders (DSM-IV), 4th Edition. Each item is rated based on the frequency of occurrence in the prior two weeks: 0 = "not at all," 1 = "several days," 2 = "more than half the days," and 3 = "nearly every day." The total score ranges from 0 to 27, with higher scores indicating

more severe depression. Examples of scale items include "Feeling down, depressed, or hopeless," as well as "Poor appetite or overeating."

PHQ-9 has been shown to have strong reliability and validity for use with students [37,38]. In our study, the Arabic translated version of the PHQ-9 was used and found to have good reliability with a Cronbach's alpha coefficient of 0.88 [39]. Our study also found good reliability of the PHQ-9 with a Cronbach's alpha coefficient of 0.901.

*Anxiety (Beck Anxiety Inventory (BAI); Beck et al., 1988).* Anxiety levels were evaluated using the Beck Anxiety Inventory (BAI), which is a 21-item self-report questionnaire that measures symptoms of anxiety [40,41]. Participants rated themselves on a 0–3 scale, with zero representing "Not at all" and three representing "Severely-It bothered me a lot." The total score ranged from 0 to 63, with higher scores indicating greater anxiety. The questionnaire covers common anxiety symptoms, such as fear of losing control, fear of dying, increased heart rate, and worry of the worse happening. The BAI has shown high internal consistency (Cronbach's alpha = 0.94) and acceptable reliability (r = 0.67) in previous research [42]. In the Arabic translated version of the 21-BAI scale, Cronbach's alpha was estimated to range between 0.83 and 0.90 [43]. In our study, the BAI scale demonstrated excellent internal consistency with a Cronbach's alpha coefficient of 0.944.

*Stress (Perceived Stress Scale (PSS); Cohen, Kamarck & Mermel-stein, 1983).* Stress was assessed using the Perceived Stress Scale (PSS), a 10-item questionnaire that evaluates stress symptoms [44]. The PSS comprises both negative and positive elements that assess lack of control, unpleasant affective reactions, and the ability to cope with current stressors. For instance, items include "How often have you felt nervous or stressed?" and "How often have you felt confident about your ability to handle your personal problems?". Participants were required to rate the frequency of their experiences over the past month on a five-point Likert scale, ranging from 0 (never) to 4 (very often). Scores on the PSS-10 range from 0 to 40, with higher scores indicating greater levels of stress. To calculate the total score, we reversed the scores on the four positive items (i.e., items 4, 5, 7, and 8).

The PSS is a reliable and valid measure of global stress that has been widely used in various settings and languages [45–48]. In this study, we found a Cronbach's alpha coefficient of 0.846 for the PSS-10 scale, indicating good internal consistency. Previous research has also reported good reliability for the Arabic version of the PSS-10 (Cronbach's alpha = 0.74) [49].

## Quality of life

The *Quality of Life Scale* (QOLS) is a 16-item instrument used to measure six conceptual domains of quality of life: material and physical well-being, relationships with other people, social, community and civic activities, personal development and fulfillment, recreation and Independence, the ability to do for yourself [50,51]. It was developed and validated in the United States [51]. It has been translated into at least 16 different languages including Arabic. The "delighted-terrible" 7-point scale was utilized. The QOLS is self-administered by filling out a questionnaire. The QOLS is calculated by adding the scores from each component to produce a total score for the instrument. The range of possible scores is from 16 to 112. The QOLS values are added together, and a higher score implies a better quality of life. Our study yielded a Cronbach alpha value of 0.911, indicating a high degree of internal consistency.

## Data analysis

All variables considered in this study were summarized using descriptive statistics. Means and standard deviations (SDs) were used to summarize continuous variables, whereas frequencies and percentages were used to summarize categorical variables. The quality of life was treated

as a continuous dependent variable. The normality of the data was evaluated using the Shapiro-Wilk normality test, a common statistical tool used to assess the normality assumption of a dataset. The QOL was modeled using four multiple linear regression models including sociodemographic, health-related, lifestyle, and mental health factors. Adjusted beta coefficient (β) was reported. The adjusted R-squared was utilized to indicate the level of goodness-of-fit for these models. The R programming language was used for the analysis (version 4.1.2). The cutoff point for statistical significance was 0.05.

## Results

Out of the initial sample of 1700 students, 329 responded, resulting in a response rate of 20%. The characteristics of the study participants' descriptive statistics are listed in Table 1. The participants' average (SD) age was 24.99 (7.39) years. Women made up the majority of the participants (63.8%). Students were enrolled in a range of university programs, with 43% of the sample being undergraduates. More than two thirds (77.5%) of participants reported monthly household incomes of no more than $450. More than half (52.3%) of the students are not in a relationship. Sixty-four percent of the respondents said that religion has a significant role in their daily life. The average number of household members was 4.43.

**Table 1. Socio-demographic factors and the association with quality of life (N = 329).**

| N<br>Mean (SD) | Quality of life | | |
|---|---|---|---|
| | 288<br>76.03 (15.6) | | |
| | | Adjusted Model | |
| | n (%) | Beta (SE) | P-value |
| **Age (Mean (SD))** | 24.99 (7.4) | 0.28 (0.19) | 0.139 |
| **Gender** | | | |
| Men | 77(23.4) | Ref | |
| women | 210(63.8) | -1.25 (2.28) | 0.585 |
| *Missing* | 42(12.8) | | |
| **University program** | | | |
| Undergraduate degree | 143(43.5) | -2.53 (2.21) | 0.253 |
| Certificate program | 14(4.3) | -8.86 (4.57) | 0.054 |
| Graduate program (MA or MSc) | 141(42.9) | Ref | |
| PhD Program | 12(3.6) | -6.90 (4.10) | 0.093 |
| MD program | 19(5.8) | -2.30 (5.10) | 0.652 |
| *Missing* | 0(0) | | |
| **Relationship status** | | | |
| Not in a relationship | 172(52.3) | Ref | |
| In a relationship | 118(35.9) | 0.40 (0.66) | 0.544 |
| *Missing* | 39(11.9) | | |
| **Importance of religion in daily decisions** | | | |
| Not important | 70(21.3) | Ref | |
| Important | 213(64.7) | 6.40 (2.33) | **0.006** |
| *Missing* | 46(14.0) | | |
| **Household Income (in Dollars)** | | | |
| 450 or less | 255(77.5) | Ref | |
| >450 | 74(22.5) | 5.25 (2.19) | **0.017** |
| **Number of household members (Mean (SD))** | 4.43 (1.6) | 0.59 (0.55) | 0.287 |
| **$R^2$** | 0.044 | | |

We used Cronbach alpha to assess the internal consistency of the QOL scale. The Cronbach alpha was 0.911 indicating a strong internal consistency. The study participants' average (SD) QOL score was 76.03 (15.6).

The adjusted associations between each socio-demographic factor and the QOL outcome are shown in Table 1. A positive relationship between QOL and the importance of religion in daily decisions was detected (β = 6.40, p = 0.006). Students who assign greater importance to religion in their daily decision-making processes tend to have higher levels of quality of life. Furthermore, results demonstrated a significant positive association between the household income and QOL (β = 5.25, p = 0.017). As the household income increases, there is a corresponding improvement in the overall quality of life.

In general, almost 60% of students rated their health as good, very good, or outstanding. The majority of students (73.6%) followed the COVID-19 prevention measures, and roughly 25% of them had the infection. Over half of the students (57.4%) sought out private counseling. Table 2 reports on the associations between health-related factors and QOL. According to adjusted findings, there is a correlation between students' general health ratings and QOL (β $_{Excellent/poor}$ = 23.52, p <0.001). Better quality of life is associated with higher self-perceived health. Additionally, students who did not receive private counseling reported higher QOL (β = 4.05, p = 0.020).

About two-thirds (63.5%) of the participants adopted a healthy diet during the pandemic, as shown in Table 3. About 12% of the respondents indicated an increase in their own cigarette and shisha smoking, as well as alcohol consumption. Nearly 50% of the students' physical activity dropped, whereas 31% of them increased. Of the participants, 32.2% slept for less than seven hours, and 17.3% slept for nine hours or more. Most of the participants (70%) used the internet for at least three hours per day (Table 3).

According to adjusted results, eating a healthy diet is positively associated with QOL (β = 4.62, p = 0.026). Students who maintained a healthy diet indicated improved QOL. The quality

**Table 2. Health related factors and the association with quality of life (N = 329).**

|  | | Adjusted Model | |
|---|---|---|---|
|  | n (%) | Beta (SE) | P-value |
| **Overall rated health** | | | |
| Poor | 29(8.8) | Ref | |
| Fair | 100(30.4) | 6.05 (3.64) | 0.116 |
| Good | 121(36.8) | 11.36 (3.67) | **0.002** |
| Very good | 66(20.1) | 20.16 (3.75) | **<0.001** |
| Excellent | 13(4.0) | 23.52 (4.85) | **<0.001** |
| **Adherence to COVID-19 preventive measures** | | | |
| No | 41(12.5) | Ref | |
| Yes | 242(73.6) | 3.69 (2.66) | 0.167 |
| *Missing* | 46(14.0) | | |
| **Infected with COVID-19** | | | |
| No | 197(59.9) | -0.20 (1.79) | 0.909 |
| Yes | 86(26.1) | Ref | |
| *Missing* | 46(14.0) | | |
| **Access to private counseling** | | | |
| No | 140 (42.6) | 4.05 (1.73) | **0.020** |
| Yes | 189 (57.4) | Ref | |
| **R$^2$** | 0.189 | | |

**Table 3. Lifestyle factors and the association with quality of life (N = 329).**

| | | Adjusted Model | |
|---|---|---|---|
| | n (%) | Beta (SE) | P-value |
| **Follow healthy diet** | | | |
| No | 120(36.5) | Ref | |
| Yes | 209(63.5) | 4.62 (2.06) | **0.026** |
| **Cigarette smoking** | | | |
| No practice | 279(84.8) | Ref | |
| Reduced | 8(2.4) | -3.78 (6.28) | 0.547 |
| Increased | 42(12.8) | -6.25 (2.86) | **0.030** |
| **Shisha smoking** | | | |
| No practice | 262(79.6) | Ref | |
| Reduced | 27(8.2) | 2.73 (2.77) | 0.324 |
| Increased | 40(12.2) | 1.33 (2.99) | 0.656 |
| **Alcohol consumption** | | | |
| No practice | 252(76.6) | -4.47 (2.8) | 0.112 |
| Reduced | 38(11.6) | Ref | |
| Increased | 39(11.9) | -6.61 (3.5) | 0.06 |
| **Physical activity** | | | |
| No practice | 63(19.1) | 1.28 (2.73) | 0.638 |
| Reduced | 164(49.8) | Ref | |
| Increased | 102(31.0) | 6.67 (1.86) | **<0.001** |
| **Sleeping hours** | | | |
| <7 | 106(32.2) | Ref | |
| 7–9 | 166(50.5) | 1.58 (1.97) | 0.423 |
| >9 | 57(17.3) | -1.96 (2.77) | 0.480 |
| **Internet use (in hours)** | | | |
| <1 | 12(3.6) | Ref | |
| [1–2] | 31(9.4) | -2.61 (3.45) | 0.45 |
| [2–3] | 59(17.9) | -1.4 (2.95) | 0.636 |
| [3–4] | 57(17.3) | -1.85 (3.02) | 0.539 |
| ≥4 | 170(51.7) | -7.01 (2.49) | **0.005** |
| **$R^2$** | 0.121 | | |

of life was likewise associated favorably with physical exercise ($\beta$ = 6.67, p <0.001). An increase in cigarette smoking was associated with a decrease in quality of life ($\beta$ increased = -6.25, p = 0.030), indicating that students who smoked cigarettes tended to have lower quality of life scores. Similarly, excessive internet use ($\beta \geq 4$ hours = -7.01, p = 0.005) showed a negative association with quality of life, suggesting that students who spent more than four hours a day on the internet tended to experience lower quality of life.

As demonstrated in Table 4, depression, anxiety, and stress had mean (SD) scores of 10.18 (6.83), 18.81 (14.42), and 21.97 (7.30), respectively. The adjusted model revealed a significant negative association between depression and quality of life ($\beta$ depression = -0.56, p = 0.002). Therefore, higher levels of depression were associated with lower quality of life scores. Similarly, the analysis indicated a negative association between stress and quality of life ($\beta$ stress = -0.93, p < 0.001). Higher levels of stress were found to be associated with decreased quality of life.

To assess the normality of the data, we conducted the Shapiro-Wilk normality test for all models. The test results revealed no significant deviation from normality for all models, with

Table 4. Mental Health factors and the association with quality of life (N = 329).

| Mental Health factors | Mean (SD) | Adjusted Model | |
|---|---|---|---|
| | | Beta (SE) | P-value |
| Depression | 10.19 (6.8) | -0.56 (0.18) | **0.002** |
| Anxiety | 18.8 (14.4) | 0.07 (0.07) | 0.314 |
| Stress | 21.9 (7.3) | -0.93 (0.15) | **<0.001** |
| $R^2$ | 0.362 | | |

p-values of 0.051, 0.539, 0.089, and 0.169 for socio-demographic, health-related, lifestyle, and mental health models, respectively, indicating that the data was normally distributed. These results are also represented in Fig 1.

## Discussion

The COVID-19 pandemic has posed multiple threats to the global population and has had a broad influence on public health by disrupting people's quality of life. The findings of this study provide valuable insights into the impact of the COVID-19 pandemic on the quality of life of university students in Lebanon. By examining various factors that influence students' well-being during these challenging times, this study contributes to the existing literature on the subject.

Our statistical analysis showed that the greater the importance of religion in daily decisions, the higher the quality of life for students. In line with previous research, the findings of this study demonstrate that religion has an important role in assisting people in coping with the COVID-19 pandemic and improving their quality of life [52,53]. A study conducted by Hu

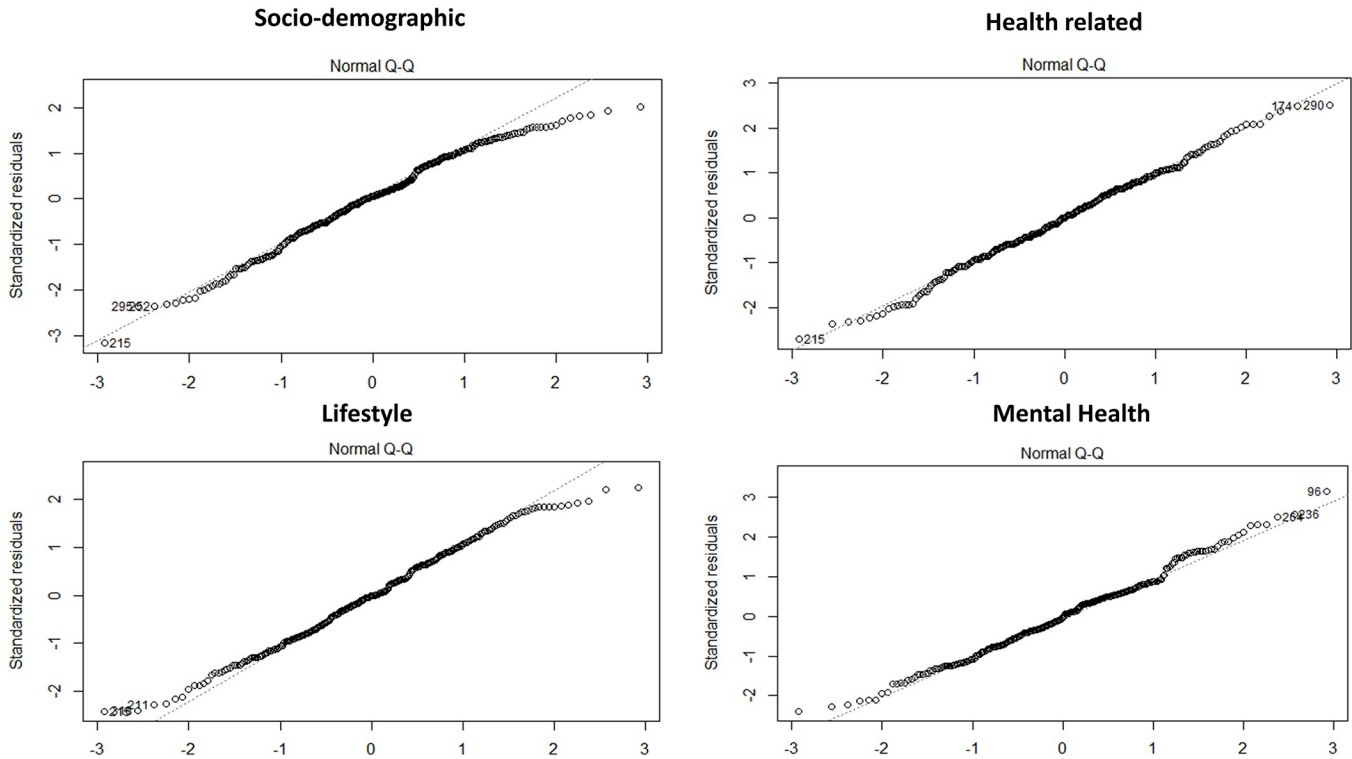

Fig 1. Normality test. Besides, all models exhibited a significant p-value (p < 0.05) indicating a satisfactory goodness-of-fit.

et al. 2021 showed that religious involvement was a significant predictor for positive quality of life among university students of different religions [54]. This is particularly true in Arab cultures, where depending on faith and engaging in religious practices are typical ways for coping with adversities.

Our study showed that healthy diet and regular exercise were significantly associated with better QOL which is in line with other studies. A study conducted among students in Canada found that individuals with higher levels of physical activity and better diet quality were significantly more likely to have higher levels of QOL than students who ate less healthily and were less active [55]. Also, according to a systematic review conducted by Wu et al. in 2019, an unhealthy eating behavior or lower food quality was linked to a decline in the health-related quality of life among children and adolescents [56]. These results draw attention to the importance of having and promoting on campus physical activities and affordable healthy food choices.

Other lifestyle factors such as cigarette smoking, and internet use were found to be negatively associated with QOL. Our findings are consistent with studies in the literature that have indicated a negative relationship between smoking and QOL among university students. [57,58]. Besides, concern about the association between excessive internet use and people's quality of life is developing in many societies. The amount of excessive Internet users among college students has reportedly increased significantly as well, particularly during the COVID-19 pandemic [59]. It was found that excessive internet use affects people's psychological and social well-being as well as their ability to function at work and in school, which lowers their quality of life overall [60]. This is consistent with our research findings, which found that heavy internet use of more than four hours per day had a detrimental impact on QOL.

Our analysis showed that students with income levels above the minimum wage have much better QOL which has been demonstrated by a previous study that showed that higher income is positively associated with health-related quality of life [60] and QOL [61] among students.

Also, our results showed that students' better quality of life is associated with higher self-perceived health. This is consistent with a recent study by Fernandes et al. 2022, which found that having a positive health self-perception improved the likelihood of having a good quality of life [62].

In terms of mental health effect, our analysis shows that depression and stress both had a negative association with QOL. This is consistent with previous findings that showed that higher severity of depressive and stress symptoms was significantly associated with lower QOL [63–66]. Depression has been shown to reduce QOL due to the mood disruption that a person with depression experiences [63]. Besides, graduate and postgraduate students at universities are in a sociodemographic age range where stress-related disorders are more common thus negatively affecting their quality of life.

Finally, our analysis showed a significant association between using counseling services and lower QOL; given the stigma of mental health and using counseling services in the country, people tend to postpone consultation until their mental health deteriorates gravely, which explains the negative association as an increased depression and stress were significantly associated with lower QOL. Campaigns destigmatizing mental health on campus as well as deployment of self-help interventions are strategies [67–69] that can mitigate this effect.

## Strength and limitations

One of the key strengths of this study is its focus on Lebanon's unique socio-economic and political context. Lebanon has been grappling with a severe economic and financial crisis, political instability, and a significant brain drain of healthcare professionals. The COVID-19

pandemic has further compounded these issues, making it crucial to understand how these intersecting factors impact the quality of life of university students. By examining sociodemographic, health-related, lifestyle, and mental health factors, this study offers a comprehensive understanding of the challenges faced by students in Lebanon during the pandemic.

Another strength of this study is its potential to inform interventions and policies aimed at improving the well-being of university students in Lebanon. The findings highlight the specific areas that require attention, such as mental health support, access to healthcare services, and addressing socio-economic disparities among students. The study's findings can guide the development of targeted support measures that consider the unique challenges faced by university students in Lebanon. Moreover, another notable strength of this study is the use of a comprehensive and validated assessment tool to measure the variables of interest. The inclusion of a valid measurement instrument enhances the accuracy and reliability of the study's findings, providing more robust evidence for the observed relationships.

However, it is important to acknowledge the limitations of this study. Firstly, the data were collected through self-report measures, which may be subject to recall bias and social desirability bias. Additionally, the study focused solely on university students, and the findings may not be generalizable to other populations or age groups. Furthermore, due to the cross-sectional design of the study, causal relationships between variables cannot be established. Longitudinal studies would be beneficial in understanding the long-term effects of the pandemic on students' quality of life.

Also, the sample included more female than male students. Conducting a multi-site study involving multiple institutions would enhance the representativeness and generalizability of the results.

## Conclusion

In conclusion, this study identifies key factors that influence the quality of life (QOL) among students. Factors that improve QOL include the importance of religion, higher household income, and maintaining a healthy diet. Conversely, higher levels of depression, stress, excessive internet use, and increased cigarette smoking negatively impact QOL. These findings highlight the significance of addressing mental health issues and promoting healthy behaviors to enhance overall well-being among students, especially during challenging times like a global pandemic. The study suggests that university administrations can take various actions, including promoting physical activities, providing affordable healthy eating options, destigmatizing mental health through campaigns, offering counseling services, and implementing self-help interventions like online mindfulness, to mitigate the impact on student QOL and support their well-being.

Future research should explore the causal relationships between these factors and QOL, as well as investigate potential mediating or confounding variables. Additionally, research focusing on diverse student populations and cultural contexts would contribute to a more comprehensive understanding of the factors influencing QOL.

## Supporting information

**S1 File.**
(CSV)

## Acknowledgments

We would like to thank the Canadian Lebanese Academic Forum for facilitating the team building effort

## Author Contributions

**Conceptualization:** Imad Bou-Hamad, Reem Hoteit, Sahar Hijazi, Dinah Ayna, Maya Romani, Christo El Morr.

**Data curation:** Imad Bou-Hamad, Reem Hoteit.

**Formal analysis:** Imad Bou-Hamad, Reem Hoteit.

**Investigation:** Reem Hoteit, Christo El Morr.

**Methodology:** Imad Bou-Hamad, Reem Hoteit, Sahar Hijazi, Dinah Ayna, Maya Romani, Christo El Morr.

**Project administration:** Christo El Morr.

**Software:** Imad Bou-Hamad, Reem Hoteit.

**Supervision:** Christo El Morr.

**Validation:** Imad Bou-Hamad, Christo El Morr.

**Writing – original draft:** Imad Bou-Hamad, Reem Hoteit, Christo El Morr.

**Writing – review & editing:** Reem Hoteit, Sahar Hijazi, Dinah Ayna, Maya Romani, Christo El Morr.

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
