## [Decision Letter · Decision Letter 0]

25 Apr 2023

PONE-D-23-02758Effects of mental health factors, lifestyle, and sociocultural characteristics on students’ quality of life in response to the COVID-19 pandemic: A cross-sectional studyPLOS ONE

Dear Dr. Imad Bou-Hamad

Thank you for submitting your manuscript to PLOS ONE. After careful consideration, we feel that it has merit but does not fully meet PLOS ONE’s publication criteria as it currently stands. Therefore, we invite you to submit a revised version of the manuscript that addresses the points raised during the review process.===============================

Major comments:The manuscript is written on a contemporary issue. However, there are a lot of confusions regarding main theme of the research including logical sequencing of the various parts of it.

- In the title you have mentioned "......in response to covid-19 pandemic", but I could not find link about the title and remaining parts of whole manuscript especially the sampling unit.

- Sample size is not clearly mentioned.

- Methodology portion should be robust, you should re-write it including all the sub heading of the methodological section.

- Discussion seems little incomplete.

- Conclusion does not linked to its result section. You cannot write anything beyond your finding in recommendation section.

We look forward to receiving your revised manuscript.

Kind regards,

Kishor Adhikari, Ph.D.

Academic Editor

PLOS ONE

2. You indicated that you had ethical approval for your study. In your Methods section, please ensure you have also stated whether you obtained consent from parents or guardians of the minors included in the study or whether the research ethics committee or IRB specifically waived the need for their consent

ACADEMIC EDITOR: 

We look forward to receiving your revised manuscript.

**Comments to the Author**

1. Is the manuscript technically sound, and do the data support the conclusions?

Reviewer #1: Partly

Reviewer #2: Partly

2. Has the statistical analysis been performed appropriately and rigorously? 

Reviewer #1: I Don't Know

Reviewer #2: No

3. Have the authors made all data underlying the findings in their manuscript fully available?

Reviewer #1: No

Reviewer #2: Yes

4. Is the manuscript presented in an intelligible fashion and written in standard English?

Reviewer #1: No

Reviewer #2: Yes

5. Review Comments to the Author

Reviewer #1: Introduction

In introduction section (literature review) insufficiently provided framing of articles but somehow try to identifies the need of the study. Study objectives are not clearly stated. However, I find lacked in appropriate gap analysis. Mostly part of the Introduction section is focused on Covid 19 without connecting outcome and predictive variables.

Methodology

Methodology section needed to written in a robust way. This is the heart of the study. Authors has tried to explain but still certain limitations were noticed. Some of the tool used had not appropriately mentioned for instances Patient Health Questionnaire (PHQ-9), Beck Anxiety Inventory (BAI-21) and Perceived Stress Scale. Some other tool used was mentioned appropriately in other hand. Please try to rectify. Though the researchers has used validated questions they must mentioned regarding the validity and the reliability of such tool in the present social context also. There is no information regarding validity, reliability of standard tools as well as the tools used for lifestyle practices and socio demographic information. It is essential to mention that how semi-structure tool has developed. No information about non response rate. This is the Web based study so chance of non-response is high. Did researcher do anything to reduce non-response? How they approaches for the participants are also not clear.

There is also not mentioned regarding pilot-testing.

Discussion: I do not feel that there is a significant novel contribution to the data shown at present, or the novelty is not clearly stipulated. Strength of the discussion section is very low. Recommend to work on improvement of it through appropriate comparison, contrasting, fostering new insights and giving direction to future research.

PLOS authors have the option to publish the peer review history of their article. If published, this will include your full peer review and any attached files. Do you want your identity to be public for this peer review? (Answer options: No)

Reviewer #2: This study attempts to measure the different factors associated with university students' quality of life. To improve the manuscript i suggest:

1. Abstract needs some attention: Authors did not mention the study design, study period and statistical significant set value in the abstract. Further , conclusive paragraph is missing. Abstract should be reviewed after clarifying the issues in the methods.

2. Introduction is inadequate: Authors need to add the paragraph explaining why they want to explore the topic explaining the status of their country/settings

3. Methods need elaboration and clarification: I believed that the purpose of the study was to investigate the social, lifestyle and mental health factors that are associated with QOL. There is no mention of the study design in the method section. Authors must describe the study setting, number of the students enrolled in the university. If the setting was well known, authors could have used the finite population formula to calculate the sample size. Specify the electronic platform through which students received a link to the survey. Authors did not address the cultural variable. Following variables were not defined: importance of religion in daily decisions, conspiracy behind covid-vaccine, adherence to COVID-19 preventive measures, private counselling. Likewise, lifestyle practices such as cigarette and alcohol intake, physical activity, follow healthy diet. What is the difference between MD program and graduate program (MA or MSc). what is the meaning of two category of household income in dollar?.

4. Data analysis needs addition: Authors did not mention the normality of the data. please include the normality test and value. why you chose linear regression? your sampling is convenience and normality of the data not mentioned. first check normality and chose the test accordingly

5. Result and discussion can only be evaluated after clarification of the issues in the study methods. Authors need to mention the response rate in the result section. Explanation of the result (i.e. this implies that the quality of life improves with income; when students increased their cigarette smoking and internet usage, there was a deterioration in their QOL; participants' quality of life was poorer when they experienced more stress and depression) not appropriate in the result section. Explanation of the results is part of the discussion. Though beta coefficient was reported, model fitness and model explanation value were not mentioned in the table 2, 3, and 4.

6. Authors need to write the conclusion based on the findings of the study because conclusions look like a recommendations

6. PLOS authors have the option to publish the peer review history of their article (what does this mean?). If published, this will include your full peer review and any attached files.

Reviewer #1: No

Reviewer #2: No

---

## [Author Response · Author response to Decision Letter 0]

21 Jun 2023

26-May-2023

PLOS ONE

Re Manuscript ID: PONE-D-23-02758 entitled “Effects of mental health factors, lifestyle, and sociocultural characteristics on students’ quality of life in response to the COVID-19 pandemic: A cross-sectional study”.

We would like to thank the editor and reviewers for the opportunity to review our manuscript entitled: “Effects of mental health factors, lifestyle, and sociocultural characteristics on students’ quality of life in response to the COVID-19 pandemic: A cross-sectional study”.

We deeply appreciate all the helpful feedback and suggestions that helped us enhance the article's content. We answered all the reviewers' comments and made significant revisions accordingly. The following were the key changes: 

1. We improved the paper (title, abstract, introduction, methodology, and discussion) to better represent the study's objectives and contribution and to eliminate any ambiguity. 

2. We strengthened the methodology (sampling procedure, scales, explanation, etc.) 

3. We refined the conclusion.

Our responses (in bold black and blue, where blue indicates newly inserted or updated statements in the paper) to the Editor and Reviewers’ comments are described below in a point-to-point manner.

Authors’ response: 

Please note that that the manuscript format has been updated to match the requirements of PLOS ONE.

Authors’ response: 

Because the study included 18-year-old undergraduate and graduate students, parental or guardian consent was not required.

Authors’ response: 

Please note that additional information on participant consent has been provided.

“The survey began with participants receiving a consent form that safeguarded their interests, provided them with relevant information regarding their rights and responsibilities, and assured the confidentiality of their information.”

“All survey participants provided their written informed consent online before completing the survey.”

Authors’ response: 

Please note that the ethics statement is now only included in the Methods section.

Authors’ response: 

Please note that once the manuscript is accepted for publication, we will provide the relevant accession numbers or DOIs necessary to access the data. In the meantime, we will upload the dataset as Supporting Information files.

Authors’ response: 

Thank you for your understanding. Please note that once the manuscript is accepted for publication, we will provide the relevant accession numbers or DOIs necessary to access the data.

Academic Editor comments:

Major comments:

The manuscript is written on a contemporary issue. However, there are a lot of confusions regarding main theme of the research including logical sequencing of the various parts of it.

- In the title you have mentioned "......in response to covid-19 pandemic", but I could not find link about the title and remaining parts of whole manuscript especially the sampling unit.

Authors’ response: 

Thank you for your comment. Please note that the title has been modified and that the phrase "in response to the COVID-19 pandemic" has been removed. 

“Coping with the COVID-19 pandemic: A cross-sectional study to investigate how mental health, lifestyle, and socio-demographic factors shape students' quality of life”

- Sample size is not clearly mentioned.

Authors’ response: 

We appreciate your feedback. We have taken your input into consideration and have made the necessary updates to clearly state the sample size in the Method, Results, and Abstract sections of the study.

“The study sample included 329 undergraduate and graduate students who were 18 years of age or older and enrolled between Spring 2020-2021 and Fall 2021-2022 at the American University of Beirut (a private university) and the Lebanese University (a public university).”

- Methodology portion should be robust, you should re-write it including all the sub heading of the methodological section.

Authors’ response: 

Thank you for your comment. We have revised the methodology section to enhance its clarity and accuracy.

- Discussion seems little incomplete.

Authors’ response: 

Thank you for your input. We have modified the discussion section and highlighted the strengths and importance of this study.

- Conclusion does not linked to its result section. You cannot write anything beyond your finding in recommendation section.

Authors’ response: 

Thank you for your valuable feedback. We have made further refinements to the conclusion section in order to more accurately represent our research findings.

“In conclusion, this study identifies key factors that positively and negatively influence the quality of life (QOL) among students. Factors that improve QOL include the importance of religion, higher household income, and maintaining a healthy diet. Conversely, higher levels of depression, stress, excessive internet use, and increased cigarette smoking negatively impact QOL. These findings highlight the significance of addressing mental health issues and promoting healthy behaviors to enhance overall well-being among students, especially during challenging times like a global pandemic. The study suggests that university administrations can take various actions, including promoting physical activities, providing affordable healthy eating options, destigmatizing mental health through campaigns, offering counseling services, and implementing self-help interventions like online mindfulness, to mitigate the impact on student QOL and support their well-being.

Future research should explore the causal relationships between these factors and QOL, as well as investigate potential mediating or confounding variables. Additionally, research focusing on diverse student populations and cultural contexts would contribute to a more comprehensive understanding of the factors influencing QOL.”

Reviewer #1: 

Introduction

In introduction section (literature review) insufficiently provided framing of articles but somehow try to identifies the need of the study. Study objectives are not clearly stated. However, I find lacked in appropriate gap analysis. Mostly part of the Introduction section is focused on Covid 19 without connecting outcome and predictive variables.

Authors’ response: 

Thank you for your feedback. We have incorporated your suggestions and made significant improvements to the introduction of our manuscript. Specifically, we have included references connecting the outcome and predictive variables to enhance the quality of our research. Additionally, we have emphasized the objectives of our study and provided a clear and concise gap analysis.

“The COVID-19 pandemic has significantly negatively affected the quality of life of university students. A study conducted by Dos Santos et al. (2022) found that scores for quality of life, stress, and depression all worsened during the pandemic compared to pre-pandemic levels (16). Additionally, the same study revealed that the pandemic led to increased physical inactivity among young people, and that female health students who lived independently and had not tested positive for COVID-19 experienced higher levels of stress during the pandemic (16). Additionally, recent research has found a decline in QoL in general and, more particularly, in the anxiety/depression domain (17, 18).

Furthermore, a study conducted by Carpi et al. (2022) revealed high rates of poor sleep quality and insomnia among university students (19), and even after controlling for the influence of health-related factors and perceived stress, this study found a substantial association between sleep quality and both physical and mental health-related quality of life (19). A study conducted in Saudi Arabia also found that smoking had a detrimental impact on the quality of life of vocational students (20). Besides, research conducted in Brazilian universities revealed that increased alcohol consumption during the COVID-19 pandemic was associated with a decline in quality of life, health satisfaction, and the meaning of life and that this negative impact was greater among women than men (21).

In Lebanon, the first verified case of COVID-19 was announced on February 21, 2020. As of 10 January 2023, the Lebanese Ministry of Public Health (MOPH) had confirmed over 1.2 million cases and over 10 700 deaths since the beginning of the COVID-19 pandemic (22). Between 2020 and 2021, the government implemented various lockdowns in an attempt to flatten the curve (3, 23). During the previous two years, all academic activities were still prohibited. These new academic norms in Lebanon have interrupted university students' daily routines and academic achievement. 

Despite the global attention given to the impact of the COVID-19 pandemic on university students' quality of life, there is a significant gap in the literature regarding the situation in Lebanon. Lebanon is a country that has experienced numerous challenges in recent years, including political instability, an economic crisis, and the COVID-19 pandemic. These factors have significantly impacted the daily lives of its citizens, including university students. While previous research has identified various factors affecting the QOL of university students, there is still a need to examine the specific impact of the pandemic and related lockdown measures on students' QOL in Lebanon. Therefore, this study aims to fill this research gap by investigating the various factors, including sociodemographic, health-related, lifestyle, and mental health factors, that affect university students' QOL in Lebanon during the pandemic. By providing insights into these factors, we hope to inform interventions and policies aimed at improving the well-being of university students in Lebanon.”

Methodology

Methodology section needed to written in a robust way. This is the heart of the study. Authors has tried to explain but still certain limitations were noticed. Some of the tool used had not appropriately mentioned for instances Patient Health Questionnaire (PHQ-9), Beck Anxiety Inventory (BAI-21) and Perceived Stress Scale. Some other tool used was mentioned appropriately in other hand. Please try to rectify. Though the researchers has used validated questions they must mentioned regarding the validity and the reliability of such tool in the present social context also. There is no information regarding validity, reliability of standard tools as well as the tools used for lifestyle practices and socio demographic information. It is essential to mention that how semi-structure tool has developed. No information about non response rate. This is the Web based study so chance of non-response is high. Did researcher do anything to reduce non-response? How they approaches for the participants are also not clear.

There is also not mentioned regarding pilot-testing.

Authors’ response: 

Thank you for your valuable feedback. We have updated the methodology section of our study to include further details on the Patient Health Questionnaire (PHQ-9), Beck Anxiety Inventory (BAI-21), and Perceived Stress Scale. Additionally, we have included information on the validity and reliability of these instruments within the context of our study. 

“Mental Health

Depression (PHQ-9; Kroenke, 2001)

The Patient Health Questionnaire (PHQ-9) (35) is a widely used and brief 9-item screening tool to detect symptoms of depression in community settings. The development of the PHQ-9 was based on the Diagnostic and Statistical Manual of Mental Disorders (DSM-IV), 4th Edition. Each item is rated based on the frequency of occurrence in the prior two weeks: 0 = "not at all," 1 = "several days," 2 = "more than half the days," and 3 = "nearly every day." The total score ranges from 0 to 27, with higher scores indicating more severe depression. Examples of scale items include "Feeling down, depressed, or hopeless," as well as "Poor appetite or overeating."

PHQ-9 has been shown to have strong reliability and validity for use with students (38, 39). In our study, the Arabic translated version of the PHQ-9 was used and found to have good reliability with a Cronbach's alpha coefficient of 0.88 (40). Our study also found good reliability of the PHQ-9 with a Cronbach's alpha coefficient of 0.901.

Anxiety (Beck Anxiety Inventory (BAI); Beck et al., 1988)

Anxiety levels were evaluated using the Beck Anxiety Inventory (BAI), which is a 21-item self-report questionnaire that measures symptoms of anxiety (41, 42). Participants rated themselves on a 0–3 scale, with zero representing "Not at all" and three representing "Severely-It bothered me a lot." The total score ranged from 0 to 63, with higher scores indicating greater anxiety. The questionnaire covers common anxiety symptoms, such as fear of losing control, fear of dying, increased heart rate, and worry of the worse happening. The BAI has shown high internal consistency (Cronbach's alpha = 0.94) and acceptable reliability (r = 0.67) in previous research (45). In the Arabic translated version of the 21-BAI scale, Cronbach's alpha was estimated to range between 0.83 and 0.90 (46). In our study, the BAI scale demonstrated excellent internal consistency with a Cronbach's alpha coefficient of 0.944.

Stress (Perceived Stress Scale (PSS); Cohen, Kamarck & Mermel-stein, 1983)

Stress was assessed using the Perceived Stress Scale (PSS), a 10-item questionnaire that evaluates stress symptoms (47). The PSS comprises both negative and positive elements that assess lack of control, unpleasant affective reactions, and the ability to cope with current stressors. For instance, items include "How often have you felt nervous or stressed?" and "How often have you felt confident about your ability to handle your personal problems?". Participants were required to rate the frequency of their experiences over the past month on a five-point Likert scale, ranging from 0 (never) to 4 (very often). Scores on the PSS-10 range from 0 to 40, with higher scores indicating greater levels of stress. To calculate the total score, we reversed the scores on the four positive items (i.e., items 4, 5, 7, and 8).

The PSS is a reliable and valid measure of global stress that has been widely used in various settings and languages (49-52). In this study, we found a Cronbach's alpha coefficient of 0.846 for the PSS-10 scale, indicating good internal consistency. Previous research has also reported good reliability for the Arabic version of the PSS-10 (Cronbach's alpha = 0.74) (53).”

Additionally, please note that we have added the response rate to the results section. And in an effort to mitigate non-response, we implemented a proactive approach by sending out several reminders to the participants. Moreover, we took great care in detailing our methodology for approaching the participants. 

An online survey was disseminated to undergraduate and graduate university students in Lebanon for this cross-sectional study. The students were provided with a link to the survey along with a detailed study description via electronic platforms such as WhatsApp and email. Two reminder messages were sent to the participants, with an interval of two weeks between them, to encourage their participation and ensure a higher response rate for the survey. At the beginning of the survey, participants were provided with a consent form that safeguarded their interests, provided them with relevant information regarding their rights and responsibilities, and assured the confidentiality of their information. Students took between 15-20 minutes to complete the survey.

As for the pilot testing, we piloted the questionnaire among the research team members to test for the timing and clarity of the questions. However, since the scales used in our study had been previously validated in various contexts, we did not require additional pilot testing. Please refer to the below references for further details on the validation of these scales.

References for BAI: 

Tafoya A., Gómez G., Ortega H., Ortiz S. Inventario de Ansiedad de Beck (BAI): Validez y confiabilidad en estudiantes que solicitan atención psiquiátrica en la UNAM. Psiquis. 2006;15:82–87. [Google Scholar]

Magan I., Sanz J., Garcia-Vera M.P. Psychometric properties of a Spanish version of the Beck anxiety inventory (BAI) in general population. Span. J. Psychol. 2008;11:626–640. doi: 10.1017/S1138741600004637. [PubMed] [CrossRef] [Google Scholar]

Antúnez Z., Vinet E.V. Escalas de depresión, ansiedad y estrés (DASS- 21): Validación de la versión abreviada en estudiantes universitarios Chilenos. Ter. Psicol. 2012;30:49–55. doi: 10.4067/S0718-48082012000300005. [CrossRef] [Google Scholar]

 References for PHQ9

Zhang, W. Q. (2020). Validity and reliability of the Patient Health Questionnaire-9 for university students (T). University of British Columbia. Retrieved from https://open.library.ubc.ca/collections/ubctheses/24/items/1.0394145

Adewuya, A. O., Ola, B. A., & Afolabi, O. O. (2006). Validity of the patient health questionnaire (PHQ-9) as a screening tool for depression amongst Nigerian university students. Journal of affective disorders, 96(1-2), 89–93. https://doi.org/10.1016/j.jad.2006.05.021

Rahman, M. A., Dhira, T. A., Sarker, A. R., & Mehareen, J. (2022). Validity and reliability of the Patient Health Questionnaire scale (PHQ-9) among university students of Bangladesh. PloS one, 17(6), e0269634. https://doi.org/10.1371/journal.pone.0269634

References for PSS

Anwer, S., Manzar, M. D., Alghadir, A. H., Salahuddin, M., & Abdul Hameed, U. (2020). Psychometric Analysis of the Perceived Stress Scale Among Healthy University Students. Neuropsychiatric disease and treatment, 16, 2389–2396. https://doi.org/10.2147/NDT.S268582

Manzar, M.D., Salahuddin, M., Peter, S. et al. Psychometric properties of the perceived stress scale in Ethiopian university students. BMC Public Health 19, 41 (2019). https://doi.org/10.1186/s12889-018-6310-z

Lee, E.-H. (2012). Review of the Psychometric Evidence of the Perceived Stress Scale. Asian Nursing Research, 6(4), 121-127. https://doi.org/https://doi.org/10.1016/j.anr.2012.08.004

Discussion: I do not feel that there is a significant novel contribution to the data shown at present, or the novelty is not clearly stipulated. Strength of the discussion section is very low. Recommend to work on improvement of it through appropriate comparison, contrasting, fostering new insights and giving direction to future research.

Authors’ response: 

Thank you for your feedback. Strengths of this study include its relevance and importance in the context of Lebanon, as well as its novelty in addressing the impact of the COVID-19 pandemic on university students' quality of life in the country. Lebanon has faced numerous challenges, including a severe economic and financial crisis, political instability, and a mass exodus of healthcare professionals. The COVID-19 pandemic has further exacerbated these existing issues, making it crucial to understand the specific effects of the pandemic and related lockdown measures on the quality of life of university students in Lebanon. This study fills a significant research gap by examining various factors, such as sociodemographic, health-related, lifestyle, and mental health factors, that influence the quality of life of university students during the pandemic in Lebanon. By providing insights into these factors, the study aims to inform interventions and policies that can improve the well-being of university students in Lebanon. The study's focus on Lebanon's unique socio-economic and political context adds value to the existing literature, contributing to a better understanding of the challenges faced by university students in this specific setting and guiding the development of targeted support measures. The novelty of this study lies in its exploration of the quality of life among university students in Lebanon, considering the intersecting factors of the pandemic, economic crisis, and political instability, which collectively make it a valuable and timely contribution to the field of research in Lebanon and beyond.

Please note that we have expanded the strengths of this study.

“One of the key strengths of this study is its focus on Lebanon's unique socio-economic and political context. Lebanon has been grappling with a severe economic and financial crisis, political instability, and a significant brain drain of healthcare professionals. The COVID-19 pandemic has further compounded these issues, making it crucial to understand how these intersecting factors impact the quality of life of university students. By examining sociodemographic, health-related, lifestyle, and mental health factors, this study offers a comprehensive understanding of the challenges faced by students in Lebanon during the pandemic.

Another strength of this study is its potential to inform interventions and policies aimed at improving the well-being of university students in Lebanon. The findings highlight the specific areas that require attention, such as mental health support, access to healthcare services, and addressing socio-economic disparities among students. The study's findings can guide the development of targeted support measures that take into account the unique challenges faced by university students in Lebanon. 

Moreover, another notable strength of this study is the use of a comprehensive and validated assessment tool to measure the variables of interest. The inclusion of a valid measurement instrument enhances the accuracy and reliability of the study's findings, providing more robust evidence for the observed relationships.

However, it is important to acknowledge the limitations of this study. Firstly, the data were collected through self-report measures, which may be subject to recall bias and social desirability bias. Additionally, the study focused solely on university students, and the findings may not be generalizable to other populations or age groups. Furthermore, due to the cross-sectional design of the study, causal relationships between variables cannot be established. Longitudinal studies would be beneficial in understanding the long-term effects of the pandemic on students' quality of life.

Also, the sample included more female than male students. Conducting a multi-site study involving multiple institutions would enhance the representativeness and generalizability of the results.”

“Future research should explore the causal relationships between these factors and QOL, as well as investigate potential mediating or confounding variables. Additionally, research focusing on diverse student populations and cultural contexts would contribute to a more comprehensive understanding of the factors influencing QOL.”

Reviewer #2: 

This study attempts to measure the different factors associated with university students' quality of life. To improve the manuscript I suggest:

1. Abstract needs some attention: Authors did not mention the study design, study period and statistical significant set value in the abstract. Further, conclusive paragraph is missing. Abstract should be reviewed after clarifying the issues in the methods.

Authors’ response: 

Thank you for your valuable feedback. To improve the abstract, we have added information regarding the study design, study period, and the statistical significance level (alpha) used in our analysis to determine the significance of our findings. Also, we added a conclusion.

Purpose

The high prevalence of COVID-19 has had an impact on the Quality of Life (QOL) of people across the world, particularly students. The purpose of this study was to investigate the social, lifestyle, and mental health aspects that are associated with QOL among university students in Lebanon.

Methods

A cross-sectional study design was implemented using a convenience sampling approach. Data collection took place between November 2021 and February 2022, involving 329 undergraduate and graduate students from private and public universities. Quality of life was assessed using the Quality of Life Scale (QOLS). Descriptive statistics, Cronbach's alpha, and Multiple Linear Regression were used to analyze the association between QOL and socio-demographic, health-related, lifestyle, and mental health factors. The significance level for statistical analysis was predetermined at α = 0.05. 

Results

The study participants' average (SD) QOL score was 76.03 (15.6) with a Cronbach alpha of 0.911. QOL was positively associated with importance of religion in daily decisions (β = 6.40, p =0.006), household income (β = 5.25, p =0.017), general health ratings (β Excellent/poor = 23.52, p <0.001), private counseling (β = 4.05, p =0.020), physical exercise (β = 6.67, p <0.001), and a healthy diet (β = 4.62, p =0.026); and negatively associated with cigarette smoking (β increased = -6.25, p =0.030), internet use (β ≥4 hours = -7.01, p =0.005), depression (β= -0.56, p =0.002) and stress (β = -0.93, p <0.001).

Conclusion

In conclusion, this study reveals the key factors that influence students' quality of life (QOL). Factors such as religion, higher income, and a healthy diet improve QOL, while depression, stress, excessive internet use, and cigarette smoking negatively impact it. Universities should prioritize initiatives like physical activity promotion, affordable nutritious options, destigmatizing mental health, counseling services, and self-help interventions to support student well-being and enhance their QOL.

2. Introduction is inadequate: Authors need to add the paragraph explaining why they want to explore the topic explaining the status of their country/settings

Authors’ response: 

Thank you for your valuable feedback. We have carefully reviewed your comments and made the necessary changes to the introduction, taking into account the unique context of our country. 

In Lebanon, the first verified case of COVID-19 was announced on February 21, 2020. As of 10 January 2023, the Lebanese Ministry of Public Health (MOPH) had confirmed over 1.2 million cases and over 10 700 deaths since the beginning of the COVID-19 pandemic (22). Between 2020 and 2021, the government implemented various lockdowns in an attempt to flatten the curve (3, 23). Besides, several modeling studies have been conducted to assess the transmission of SARS-CoV-2 and evaluate the impact of COVID-19 vaccination (24-26). One study, conducted in April 2021, predicted a significant increase in cases and deaths if schools and universities reopened, particularly considering the low vaccination rates of below 4% (26). 

Despite various efforts to encourage vaccination, such as the national COVID-19 vaccination plan aiming to vaccinate 70% of the population by the end of 2022, Lebanon has fallen far short of achieving this goal by 2023 (27, 28). As of April 8, 2023, the first-dose vaccine coverage stood at 50.4%, with second-dose coverage at 44.4%. Furthermore, only 27.6% of individuals who received the second dose went on to receive the third dose (29). As of April 8, 2023, new COVID-19 infections continue to occur in Lebanon, with an average of 100 cases reported daily. 

Despite the global attention given to the impact of the COVID-19 pandemic on university students' quality of life, there is a significant gap in the literature regarding the situation in Lebanon. The current situation is unparalleled, as the country is grappling with an overwhelming crisis characterized by deteriorating financial and political stability that has plagued Lebanon for decades, leading to a mass exodus of physicians and nurses (30, 31). It is noteworthy that the COVID-19 pandemic has exacerbated Lebanon's existing severe economic and financial crisis, which began in October 2019 (28). In March 2023, the Lebanese lira experienced a devaluation of over 95%, making it one of the most severe economic and financial crises witnessed globally since the mid-19th century, according to the World Bank (32, 33).

These factors have significantly impacted the daily lives of its citizens, including university students. While previous research has identified various factors affecting the QOL of university students, there is still a need to examine the specific impact of the pandemic and related lockdown measures on students' QOL in Lebanon. Therefore, this study aims to fill this research gap by investigating the various factors, including sociodemographic, health-related, lifestyle, and mental health factors, that affect university students' QOL in Lebanon during the pandemic. By providing insights into these factors, we hope to inform interventions and policies aimed at improving the well-being of university students in Lebanon.

3. Methods need elaboration and clarification: I believed that the purpose of the study was to investigate the social, lifestyle and mental health factors that are associated with QOL. There is no mention of the study design in the method section. Authors must describe the study setting, number of the students enrolled in the university. If the setting was well known, authors could have used the finite population formula to calculate the sample size. Specify the electronic platform through which students received a link to the survey. Authors did not address the cultural variable. Following variables were not defined: importance of religion in daily decisions, conspiracy behind covid-vaccine, adherence to COVID-19 preventive measures, private counselling. Likewise, lifestyle practices such as cigarette and alcohol intake, physical activity, follow healthy diet. What is the difference between MD program and graduate program (MA or MSc). what is the meaning of two category of household income in dollar?.

Authors’ response: 

We appreciate your feedback. In the Methods section, we have provided clarification regarding the study design and setting. It is important to note that during the data collection period, the Lebanese university faced various challenges, including faculty strikes resulting from the situation in Lebanon. Consequently, the study population was limited to a small number of faculties that were not affected by the strikes. This restricted the potential sample size and necessitated data gathering from a subset of faculties rather than the entire university population.

The electronic platforms were specified in the manuscript. 

“The students were provided with a link to the survey along with a detailed study description via electronic platforms such as WhatsApp and email.”

We have replaced the term "cultural variables" with "socio-demographic variables" in the manuscript. While the inclusion of religion prompted the use of the term "culture," we acknowledge that "socio-demographic variables" is a more comprehensive and representative term.

Also, please note that the items’ definitions are found in the socio-demographics and the lifestyle paragraph in the “METHOD/Survey Instrument” paragraph.

Socio demographics

The “importance of religion in daily decisions”, “conspiracy behind COVID-vaccine” “adherence to COVID-19 preventive measures” were defined in the Socio-demographics paragraph as follows:

“importance of religion in daily decisions (binary: not important; important), conspiracy behind COVID virus/vaccine (categorical: disapprove, neither approve nor disapprove and approve) adherence to COVID-19 preventive measures (binary: no; yes)”

We updated the socio-demographics to give more clarity to the item “private counselling”; instead of “private counseling (binary: no; yes)” the text now reads “access to private counseling (binary: no; yes)”

Lifestyle practices

The three remaining items are lifestyle items: 

- cigarette and alcohol intake

- physical activity

- follow healthy diet

They were all defined in the lifestyle practices paragraph in the “Methods/survey instrument” paragraph as follows:

“cigarette and shisha smoking (categorical: no practice, reduced and increased), alcohol intake (categorical: no practice, reduced and increased)… follow a healthy diet (binary: no; yes)”

Regarding the difference between MD and (MA or MSc):

An MD program, also known as a Doctor of Medicine program, is a professional degree program that prepares individuals to become medical doctors or physicians; whereas, a graduate program, such as a Master of Arts (MA) or Master of Science (MSc) program, is an academic degree program that focuses on advanced study and research in a specific field of study.

Please note that we have clarified the meaning of these terms in the manuscript.

“Graduate program (Master of Arts (MA) or Master of Science (MSc)), PhD Program and Doctor of Medicine program (MD)”

Regarding the household income in USD:

It is important to clarify that the household income reported in this study is in United States Dollars (USD) and is treated as a binary variable. Specifically, households are categorized as either having an income below or equal to USD 450, or having an income greater than USD 450.

The household income (binary: ≤ USD 450; >USD 450)

4. Data analysis needs addition: Authors did not mention the normality of the data. please include the normality test and value. why you chose linear regression? your sampling is convenience and normality of the data not mentioned. first check normality and chose the test accordingly

Authors’ response: 

Thank you for your suggestion. We have conducted the Shapiro-Wilk normality test, a commonly used statistical tool, to assess the normality assumption of the dataset. We have also included the respective p-values for each of the four models (Socio-demographic: 0.051, Health-related: 0.539, Lifestyle: 0.089, and Mental health: 0.169) in the manuscript, which is proof for the rationale behind our choice of linear regression. Furthermore, we have provided a graphical representation of the normality of the data for all four models in Figure 1. 

“Besides, the normality of the data was evaluated using the Shapiro-Wilk normality test, a common statistical tool used to assess the normality assumption of a dataset.”

“To assess the normality of the data, we conducted the Shapiro-Wilk normality test for all models. The test results revealed no significant deviation from normality for all models, with p-values of 0.051, 0.539, 0.089, and 0.169 for socio-demographic, health-related, lifestyle, and mental health models, respectively, indicating that the data was normally distributed. These results are also represented in Figure 1.”

5. Result and discussion can only be evaluated after clarification of the issues in the study methods. Authors need to mention the response rate in the result section. Explanation of the result (i.e. this implies that the quality of life improves with income; when students increased their cigarette smoking and internet usage, there was a deterioration in their QOL; participants' quality of life was poorer when they experienced more stress and depression) not appropriate in the result section. Explanation of the results is part of the discussion. Though beta coefficient was reported, model fitness and model explanation value were not mentioned in the table 2, 3, and 4.

Authors’ response: 

Thank you for your feedback. Please note that we have added the response rate to the manuscript.

Also, we utilized the adjusted R-squared measure to indicate the level of goodness-of fit for our linear regression models, and we have included the resulting values in the relevant tables (Table 1, 2, 3, and 4). Notably, all models exhibited a significant p-value (p < 0.05) indicating a satisfactory goodness-of-fit. 

“The adjusted R-squared was utilized to indicate the level of goodness-of-fit for these models.”

“Besides, all models exhibited a significant p-value (p < 0.05) indicating a satisfactory goodness-of-fit.”

Additionally, please note that the explanation for the results has been revised.

Students who assign greater importance to religion in their daily decision-making processes tend to have higher levels of quality of life. 

As the household income increases, there is a corresponding improvement in the overall quality of life.

An increase in cigarette smoking was associated with a decrease in quality of life (β increased = -6.25, p = 0.030), indicating that students who smoked cigarettes tended to have lower quality of life scores. Similarly, excessive internet use (β ≥4 hours = -7.01, p = 0.005) showed a negative association with quality of life, suggesting that students who spent more than four hours a day on the internet tended to experience lower quality of life.

The adjusted model (Table 4) revealed a significant negative association between depression and quality of life (β depression = -0.56, p = 0.002). Therefore, higher levels of depression were associated with lower quality of life scores. Similarly, the analysis indicated a negative association between stress and quality of life (β stress = -0.93, p < 0.001). Higher levels of stress were found to be associated with decreased quality of life.

6. Authors need to write the conclusion based on the findings of the study because conclusions look like a recommendations

Authors’ response: 

Thank you for your feedback. Please note that we have refined the conclusion based on the findings.

In conclusion, this study identifies key factors that positively and negatively influence the quality of life (QOL) among students. Factors that improve QOL include the importance of religion, higher household income, and maintaining a healthy diet. Conversely, higher levels of depression, stress, excessive internet use, and increased cigarette smoking negatively impact QOL. These findings highlight the significance of addressing mental health issues and promoting healthy behaviors to enhance overall well-being among students, especially during challenging times like a global pandemic. The study suggests that university administrations can take various actions, including promoting physical activities, providing affordable healthy eating options, destigmatizing mental health through campaigns, offering counseling services, and implementing self-help interventions like online mindfulness, to mitigate the impact on student QOL and support their well-being.

Future research should explore the causal relationships between these factors and QOL, as well as investigate potential mediating or confounding variables. Additionally, research focusing on diverse student populations and cultural contexts would contribute to a more comprehensive understanding of the factors influencing QOL.

---

## [Editor Report · Decision Letter 1]

26 Jun 2023

Coping with the COVID-19 pandemic: A cross-sectional study to investigate how mental health, lifestyle, and socio-demographic factors shape students' quality of life

PONE-D-23-02758R1

Dear Dr. Bou-Hamad,

We’re pleased to inform you that your manuscript has been judged scientifically suitable for publication and will be formally accepted for publication once it meets all outstanding technical requirements.

Kind regards,

Kishor Adhikari, Ph.D.

Academic Editor

PLOS ONE
---

## [Editor Report · Acceptance letter]

12 Jul 2023

PONE-D-23-02758R1 

Coping with the COVID-19 pandemic: A cross-sectional study to investigate how mental health, lifestyle, and socio-demographic factors shape students' quality of life 

Dear Dr. Bou-Hamad:

I'm pleased to inform you that your manuscript has been deemed suitable for publication in PLOS ONE. Congratulations! Your manuscript is now with our production department. 

Kind regards, 

on behalf of

Dr. Kishor Adhikari 

Academic Editor

PLOS ONE